# Bank Interest Rate Margin, Portfolio Composition and Institutional Constraints

## Li Xian Liu [1,*] and Milind Sathye [2]

1   College of Business, Law & Governance, James Cook University, Townsville 4811, Australia
2   Faculty of Business, Government & Law, University of Canberra, Canberra 2601, Australia
*   Correspondence: li.liu1@jcu.edu.au

**Abstract:** This study empirically examines how the bank specific factors, macro-economic, and institutional variables impact interest margins in China's banking sector. A panel data analysis of bank data for the period 1988–2015 was carried out. We found a significant association between credit quality, risk aversion, liquidity risk, and the proportion of corporate and industrial loans and the adjusted interest spread (AIS). GDP growth rate, inflation, and the proportion of national savings to the GDP were found to have significant association with the AIS. Furthermore, institutional variables were found to have a significant moderating effect on the AIS. We contribute to the literature by examining a unique context and a more accurate measure of bank interest margin not used in prior studies.

**Keywords:** adjusted interest spread; corporate and industrial loans; financial freedom; monetary freedom; government spending; economic freedom

## 1. Introduction

What are the determinants of interest rate margin in Chinese banks? This is the question that we address in the present paper. Net interest margin (NIM) refers to the difference between the lending and deposit rates of banks (Birchwood et al. 2017).

A study of interest margins is important because banks are dominant players in any economy and particularly, in developing/emerging countries, as banks are the main suppliers of finance. Bank net interest margin is also used as one of the prime indicators of competitiveness in the financial system (Murray Review 2014). Furthermore, interest margins are regularly watched by the central banks across the world. The Federal Reserve and the Reserve Bank of Australia (RBA), for example, publish quarterly charts of bank interest margins (recent years have seen a general decline in bank interest margins in many countries). Interest margins are also important from the perspective of savings and investments as high interest margins can adversely impact these, particularly in developing economies where capital markets are not fully developed (Ben Khediri and Ben-Khedhiri 2009).

The Chinese context is important because of the dominance of state-owned banks (SOBs), a feature not seen in western economies, and as such presents unique issues not considered in prior literature that focused on developed countries. Second, the financial stability risks in China remain elevated. 'The level of debt in China has risen significantly over the past decade to reach very high levels, with particularly strong growth in lending from the less regulated and more opaque parts of China's financial system' (RBA  Reserve Bank of Australia, p. 1). Third, the technology usage in banking in an emerging economy such as China is different from that of developed countries with mature technology. This can influence operating costs—one of the determinants identified in prior studies. China embarked on banking reforms, among others, to improve banking operations technology (Tan 2016) and reduce operating costs. Lastly, as per RBA  (Reserve Bank of Australia), the interest margins in the US declined

from nearly 4% (2000) to about 2.5% (2016), those of UK from close to 2% to 1%, Japan over 1% to less than 1%, and Australia from 3.7% to 2.3% in the same period. However, the bank interest margins in China which were close to 2.2% (2004), rose to about 3.7% (2014) during the study period (though stood at 2.7% in 2016). The interest margins in China continue to remain high compared to other countries as above. Accordingly, China presents a unique case. As the banks in China largely depend on revenues from lending, the net interest margin is a crucial number (John 2017). The pressure on interest margins in China can be traced to tightening of regulations by China Banking Regulatory Commission (CBRC) which required that banks fund two-thirds of their activities with stable deposits, tightened loan classifications, and mandated additional risk disclosures. The lack of competition from foreign banks incorporated in China—due to policy limitations—also impacted the interest margins.

Against the above background, a study of the determinants of bank interest margins (as measured by the adjusted interest spread—AIS) in China assumes importance. To answer the research question indicated above, we examined the Chinese local and foreign banks data for the years 1988–2015. The fixed effect as well as the random effect models were tested. The study found that of the bank specific variables, such as credit quality, risk aversion, liquidity risk, and the proportion of corporate and industrial loans, had significant association with the AIS. The macro-economic variables such as the GDP growth rate, inflation, and the proportion of national savings to the GDP exerted significant influence on the AIS. None of the institutional distance variables were found to be associated with the AIS, though these were found to have some significant moderating effects on the AIS.

The present study contributes to the extant literature in several ways. First, the study provides evidence from the contextual uniqueness of China. Second, SOBs in China have access to cheap government equity in contrast to the private equity of banks in western countries which can impact interest margins as large non-interest-bearing funds (that is, government business) becomes available to them. Consequently, the interest margin determination models applicable in western countries may not be relevant in the Chinese context. Third, prior studies have used interest margin as the dependent variable but there are differences between interest margin, spread, and the adjusted spread, though the concepts are related (RBA Reserve Bank of Australia). 'The difference between the measures will be larger if there are substantial non-interest-bearing deposits; in this case the spread will significantly underestimate the true difference between the average interest received and paid' (RBA Reserve Bank of Australia, p. 1). Consequently, the RBA uses the AIS as the measure instead of the simple interest margin. The present study uses this more accurate measure as the dependent variable in contrast to prior studies.

The rest of the paper proceeds as follows: Section 2 introduces the institutional environment as well as the interest rate environment in which the banks in China—whether local or foreign—operate. Section 3 presents a review of literature and describes the empirical model for interest margin determination used by us. Section 4 is about the data and method, Section 5 presents the empirical results and Section 6 concludes.

## 2. Chinese Banking Industry: Institutional Setting and Interest Rate Liberalization

The banking industry, as a major financial intermediary and a resource allocator in Chinese economy, has been undergoing substantial reforms over the last four decades. Despite these financial sector reforms, however, the Chinese financial market continues to be heavily regulated with limited links to the larger economy (Allen et al. 2017).

Starting with a planned economy and a planned interest rate regime, China later adopted a gradual and cautious approach to liberalize interest rates. Though the central bank—the People's Bank of China (PBOC)—was established in 1984, deregulation of bank lending and deposit rates formally started after 1999 when the PBOC removed all restrictions on money market and bond market rates. In October 2015, the PBOC removed the deposit interest rate ceiling, paving the way for the full liberalization of interest rates. However, the PBOC will continue setting benchmark savings and lending rates for an unspecified period. So far, the Chinese banking system has largely focused on traditional financial

intermediation between savers and borrowers. Consistent with this, around two-thirds of Chinese banks' income is generated by these activities (CBRC Chinese Banking Regulatory Commission). Further, Chinese banks command enough capital resources and have been able to transform most of the nation's vast savings into deposits, and then into loans, most of which go to state-owned-enterprises (SOEs) (Liang 2016). The five largest state controlled commercial banks in China account for around one-half of Chinese banking system assets, and deposits contributed 85% of loans to SOEs in 2009 (Grant et al. 2012; Cary 2013).

Although the PBOC set the benchmark interest for deposit and lending activities, the actual rates set by individual banks could float within a small range around the benchmark rate. After 2000, the gap between the benchmark deposit rate and lending rate was almost fixed. Under these circumstances, the interest rate margins for the Chinese banks are protected. Banks can lock their profits for each dollar they lend due to this theoretically fixed interest margin. Consequently, prima facie, there would be little interest in studying bank interest margins in China. However, within the limits set by the PBOC, many factors do impact bank interest margins. What are these factors then? Do factors like size, proportion of non-performing loans, composition of assets and liabilities and others impact Chinese bank interest margins?

Traditionally, the SOEs in China, whose loan requirement is very large, are supported by SOBs (Brandt and Zhu 2000). During the transition period, China adopted a dual interest rate system where under the SOEs can borrow at subsided rates from SOBs while the non-SOEs must pay the market interest rates (which is usually higher) to finance their investment projects (Chen 2002). The extent of the subsidy received by the SOEs can be gauged from the following. The borrowing rate for SOEs was about 2% while the average annual borrowing rates for SOEs would be 3.5% without state support based solely on their stand-alone profiles (The Economist 2016). It is also reported that SOEs account for over a third of domestic investment, and local governments account for 90% of all domestic fixed-asset investment in infrastructure. One of the major funding sources of the local government are bank loans (Allen et al. 2017). However, such loans also contribute to the high level of non-performing loans (NPLs) in the Chinese banking system.

Foreign banks were permitted to operate in China after the banking sector reforms with the expectation that their presence would help enhance competitive efficiency and improve the structure of the country's banking system (Leung 1997). According to the annual report issued by CBRC (Chinese Banking Regulatory Commission) by the end of 2015, there were 37 wholly foreign-owned banks (with 306 branches under them), two joint-venture banks (with four branches under them), and one wholly foreign-owned finance company operating in China. However, there are some restrictions on the expansion of foreign banks. During the 1990s, foreign banks could only provide foreign currency loans and deposits to firms with foreign investment or to foreign individuals living in China. In December 2006, the Chinese government removed the geographical, clients, and currency restrictions on foreign banks and thereby sought to establish a level-playing field between foreign and local banks. Foreign banks, however, continue to face challenges as they are expected to provide credit for purposes such as agriculture and rural credit, or finance for small and medium enterprise (SMEs). However, such type of lending involves enormous risk especially because foreign banks may not have required insights in the local situation and cultural issues involved. As a result of these factors, though China has the largest banking sector in the world, foreign banks accounted for only 1.38% of the total banking assets in China as of December 2015 (a 0.35% decline compared to their market share in 2013). The lack of competition from foreign banks naturally influences overall bank interest margins.

## 3. Literature Review

Ho and Saunders (1981) seminal paper on the determinants of interest margin elucidates the theory and provides the evidence. The study integrated the expected utility theory and the hedging theory and drew on the literature on the bid-ask prices for security market dealers. The study found that the degree of bank's risk aversion, the market structure in which the bank operates (that is, competition),

the average size of bank transactions, and the variance of interest rates influence bank net interest margin. They found that even in highly competitive markets, interest margins can exist because of uncertainty in transaction which they call pure spread.

Following the work of Ho and Saunders (1981), many studies on bank interest margin determination were undertaken across the world, especially in Europe and North America. Angbazo (1997) found that interest margins are positively related to core capital, non-interest-bearing reserves, and management quality, but negatively related to liquidity. The Saunders and Schumacher (2000) seven-country study (did not include China) found that the regulatory components in the form of interest-rate restrictions on deposits, reserve requirements, and capital-to-asset ratios were the determinants. Maudos and de Guevara (2004) in the context of European banking found that market power, interest risk, credit risk, risk aversion, implicit payments, and operating costs were the determinants. Lepetit et al. (2008) in the study of 12 European countries found that interest margins were highly impacted by fee income.

Claeys and Vennet (2008) presented a comparative analysis of the determinants of the bank interest margins in the Central and Eastern European countries and the Western European countries. These authors found, among others, that the presence of foreign banks reduces bank interest margins. Doliente (2005) examined determinants of net interest margins of banks in four South East Asian countries—Indonesia, Thailand, Philippines, and Malaysia—and found that bank specific factors (collateral, liquid assets, loan quality, operating expenses, and capital) were the determinants of bank interest margins in these countries. In their study of determinants of bank interest margins in Russia, Fungáčová and Poghosyan (2011) found that these were impacted by ownership type.

Other studies on bank interest margin identified factors such as interest rate volatility (Angbazo 1997; Saunders and Schumacher 2000; Carbó-Valverde and Fernández 2007; Entrop et al. 2015), credit risk (Angbazo 1997; Maudos and de Guevara 2004; Hawtrey and Liang 2008), operating costs (Williams 2007), and market power (Gischer and Juttner 2003; Williams 2007). These researchers found a positive association between the above factors and interest margin. Some studies such as Demirgüç-Kunt and Huizinga (2004) and Carbó-Valverde and Fernández (2007), however, found that the association between market power and interest margin could be negative depending upon the state of institutional development of a country.

There is limited literature on bank interest margins in China. Zhou and Wong (2008) examined the determinants of net interest margins of 81 Mainland Chinese commercial banks for the period 1996 to 2003 and found that market competition, average-operating costs, degree of risk aversion, transaction size, implicit interest payments, opportunity cost of reserve, and management efficiency were the determinants of net interest margin of banks in China. Further, the study also examined the impact of economic freedom on the overall economic wellbeing (though not on the banking sector). Zhou et al. (2008) found that net interest margin of Chinese bank was impacted by interest rate liberalization, capital adequacy norms, and regional operating restrictions. The Qi and Yang (2016) study found that foreign bank presence and short-term funding exercised negative impact on Chinese bank interest margin. García-Herrero et al. (2009) examined profitability of Chinese banks and found that better capitalized banks were more profitable. It was also found that the four SOBs were a drag on the overall banking system profitability. Sufian (2009) studied the profitability of Chinese commercial banks and found that it is positively associated with size, credit risk, and capitalization but has negative association with liquidity, overhead costs, and network embeddedness.

We improve upon these studies in several ways: (a) we use the most complete data (1988 to 2015) of Chinese and foreign banks operating in China delete full stop (b) we use a more relevant measure of dependent variable, the AIS, to more appropriately capture the reality, and (c) we consider the influence that institutional factors as well as economic factors exercise on the AIS in respect of both domestic and foreign banks which has not been considered in prior studies.

Following from the above literature, the general hypothesis is that the AIS is affected by a set of factors as presented below:

AIS = f(bank specific factors; industry and macro-economic factors; institutional factors)

## 4. Variables, Data, and Method

### 4.1. Dependent Variable

Following from the RBA (Reserve Bank of Australia), we used the AIS as a measure of the dependent variable as below:

$$Adjusted\ Interest\ Spread\ (AIS) = \frac{Interest\ income}{interest-earning\ assets} - \frac{Interest\ paid}{Total\ deposit}. \tag{1}$$

### 4.2. Independent Variables

The independent variables include bank specific factors, industry and macroeconomic variables, and institutional distance variables.

#### 4.2.1. Bank Specific Factors

Bank specific variables as below have been frequently used in the literature.

*Size of operations*: Compared to smaller clients or transactions, larger transactions reduce the frequency of operations and spread administrative overheads across a larger base, which reduces a bank's operating expenses per dollar of revenue, hence, economies of scale exist (Hawtrey and Liang 2008). Chinese SOBs can offer credit at narrower margins to large corporate clients such as the Chinese SOEs, for example, than what the smaller banks can offer. Accordingly, the logarithm of the volume of loans as the measure of size of operation is employed in the present study to capture the scale effects.

*Credit Quality*: Credit quality (risk) is measured by the proportion of loan loss provisions to total loans indicating bank's credit quality (Dietrich and Wanzenried 2011; Schwaiger and Liebig 2009). A higher ratio indicates a lower credit quality and therefore exercises a negative influence on the AIS.

*Risk aversion*: Banks that are more risk-averse will charge higher margins (Maudos and de Guevara 2004). Following from McShane and Sharpe (1985) and Maudos and de Guevara (2004) we used the ratio of shareholders' funds to total assets as a measure of risk aversion.

*Loan quality*: The loan quality (as a measure of credit risk) is calculated as the proportion of non-performing loans to gross loans. In theory, banks with higher credit risk are likely to adjust upwards their interest margins to cover the potential losses (Angbazo 1997; Mody and Peria 2004; Hawtrey and Liang 2008).

*Operational inefficiency*: Higher operational ratio implies higher interest margins and is indicative of firm's operational inefficiency. Following from Saunders and Schumacher (2000), Mody and Peria (2004), Maudos and de Guevara (2004), Zhou and Wong (2008), we included the operational inefficiency variable as a control variable in the bank interest margin modelling. We measured the operational inefficiency as the overhead expenses to average assets.

*Liquidity risk*: In theory, holding higher proportion of liquid assets involves opportunity cost. Banks would pass this cost on to the borrowers and depositors. As a result, these costs need to be priced into the setting of interest margin (Kashyap and Stein 1995).

*Proportion of corporate and industrial loans*: is proxied by the proportion of corporate and industrial loans to total loans to capture the effect of portfolio composition on the AIS. The highly profitable corporate loans segment is open for fierce competition and impacts spread. 'Vigorous competition for new large corporate loans is being induced by the narrow spreads available on market-based funding, as well as the growing presence of a number of foreign banks, particularly Asian-owned banks, in the

Australian business loan market' (RBA Reserve Bank of Australia, p. 37). Memmel (2014) asserts that bank's portfolio composition has a huge impact on net interest margins.

### 4.2.2. Industry and Macroeconomic Variables

Market share represents market power. It is measured as the proportion of bank assets to the total assets of the banking sector in the country. Higher market share represents higher market power.

The Hirschman–Herfindhal index (HHI) is used to calculate the degree of concentration in the banking industry. The market concentration indices exhibit the general form as below:

$$HHI_i = \sum_{i=1}^{n} s_i w_i, \tag{2}$$

where $HHI_i$ is the market concentration index for bank $i$, $s_i$ is the market share of bank $i$, $w_i$ is the weight attached to the market share, and $n$ is the number of banks in the market in question.

Macroeconomic environment is captured by the overall economic growth which is measured by GDP growth rate, and GDP deflator. Further, being one measure of domestic investment, national savings to GDP has important implications for the economy as well.

### 4.2.3. Institutional Variables

Four institutional distance variables are included in the study which are financial freedom distance, government spending distance, monetary freedom distance and government effectiveness distance. Economic freedom is an integrated index. One drawback is that a single measure may not be able to properly capture the overall economic environment faced by the bank. Furthermore, a highly aggregated index makes it difficult to draw policy conclusions. Therefore, we chose three sub-categories that are related to bank operations and investigate their effects on the AIS. These variables are based on Fraser Institute (2017) economic freedom of the world annual reports, for the years under study, which measure a country's openness to the other world. Financial freedom measures banking efficiency as well as independence from government control or interference in the financial sector. State ownership of banks is considered as a burden that adds to operational inefficiency. Government spending indicates the level of government contribution in the economy, and monetary freedom represents the price stability and liberalization. Each of these categories is graded on a scale of 0 to 100 with higher values corresponding to better outcomes. It is important to note that this study uses freedom distance values rather than merely the index per se as there are 39 foreign banks operating in China. 'Government effectiveness captures perceptions of the quality of public services, the quality of the civil service and the degree of its independence from political pressures, the quality of policy formulation and implementation, and the credibility of the government's commitment to such policies' (World Bank 2019).

Foreign and domestic banks differ in their management strategies, clients, knowledge of the local market, international regulatory arbitrage, and international business platform. They face different kind of competitiveness in the banking market and different advantages in business operations (Elyasiani and Rezvanian 2002), which eventually affect their interest margins.

Institutional distance refers to the relative distance between China and the home country of the foreign bank. It is calculated in the same way as the cultural distance is computed. Distance calculation, illustrated by cultural distance (Kogut and Singh 1988) is shown below:

$$ID_j = \sum_{i=1}^{4} \frac{\{\frac{(I_{ij} - I_{iO})^2}{V_i}\}}{4}, \tag{3}$$

where $ID_j$ is the institutional distance between host country $j$ and the other country, $I_{ij}$ is country $j$'s score on the $i$th institution dimension, $I_{iO}$ is the score of the other country on this dimension, and $V_i$ is the variance of the score of the dimension.

Finally, we included foreign banks in our dataset although the market share of these banks is very small in China. Empirically and theoretically, foreign banks have their competitive advantages compared to the local banks, for example, large asset base from the parent bank, international branch network, easy access to euro-currencies market, modern banking technology, and credit management practice etc. Foreign banks have more than doubled profits in 2011 to RMB 16.73 billion. They also expect to grow revenues by 20% over the next three years (PWC 2012). However, the expansion of foreign banks is limited by relevant laws and regulations of the supervisory and regulatory institutions of China.

Table 1 provides a description of all variables.

**Table 1.** Description of variables.

| Dependent Variable | Measure and Description |
| --- | --- |
| AIS | Measured as per formula indicated in this paper. Adjusted interest spread |
| **Bank specific variable** | |
| Size of Operations | Logarithm of total gross loans |
| Credit Quality | Provisional loan loss/total loans |
| Risk Aversion | Total equity/total assets |
| Loan Quality | Impaired loans (NPL)/gross loans |
| Operational Inefficiency | Overheads/average assets |
| Liquidity Risk | Liquid assets/the sum of deposits and short-term funding |
| Proportion of Commercial and Industrial Loans | |
| Local or Foreign Bank | Dummy variable that takes value of 1 for foreign banks, otherwise 0 |
| **Portfolio composition variable** | |
| Proportion of Corporate and Industrial Loans | Corporate and commercial loans/gross loans |
| **Industry and macro-economic variable** | |
| Market Concentration | Herfindahl-Hirschman Index (HHI) index (total assets) |
| Market Share | Bank assets/total banking assets in economy |
| Economic Growth | Real GDP growth rate |
| Real Interest Rate | Real Interest Rate |
| Inflation | CPI growth rate, measurement of percentage change in consumer price index |
| National Savings | Proportion of national gross savings to GDP |
| **Institutional variable** | |
| Financial Freedom Distance | The difference between the host country's (China) financial freedom index to the other 14 foreign countries and districts |
| Government Spending Distance | The difference between the host country's (China) government spending level and the other 14 foreign countries and districts |
| Monetary Freedom Distance | The difference between the host country (China) monetary freedom index and the other 14 foreign countries and districts |
| Government Effectiveness Distance | The difference between the host country (China) government effectiveness index and the other 14 foreign countries and districts |

### 4.3. Data

Our panel dataset consists of annual data for 192 banks over the period 1998–2015. This includes five large SOBs, 33 joint-stock commercial banks, 79 urban commercial banks, 32 rural commercial banks, 40 foreign banks, and three rural cooperative banks in six groups of banks. We excluded the central bank and the three policy banks from our sample. The main data source was Bankscope from Bureau van Dijk, which compiles data mostly from the balance sheet and income statement. Industry and macroeconomic variables were obtained from the website of China Banking Regulatory Commission and the World Bank database such as the World Bank World Development indicators (WDI) database (World Bank 2017). The institutional variables were obtained from the Index of Economic Freedom report maintained by the Heritage Foundation (2018).

Furthermore, we conducted a missing value analysis and identified variables that have many missing values. We omitted the cases that had common missing values across all the variables. With the remaining cases, the percentage of missing values across variables ranged from 1.3% to 10%, and we

then conducted an imputation procedure with SPSS to systematically replace the missing values for all missing variables. In order to avoid possible effect resulting from differences in number of observations (sample size), we chose the lowest number of observations as the sample size for subsequent analyses, and deleted the rest of the cases. This left 1206 observations for all the variables in all the models.

*4.4. Methodology*

To choose the best method of conducting data analysis, a few diagnostic tests were employed. We used the Breusch–Pagan Lagrange multiplier (LM) test to choose between a random effects regression and simple OLS regression. The Prob > chi2 equals zero and as such there is evidence of significant differences across banks. Consequently, the random effects regression was run. To decide between fixed or random effects regression, the Hausman test was employed. It suggested that all models reject the null hypothesis (with Prob > chi2 = 0.000) which indicated that the fixed effect model is appropriate. The variable of Local or Foreign Bank was omitted due to collinearity, though it is an important variable we would like to investigate. Consequently, we considered Local or Foreign Bank as a moderating variable. The model is stated as below:

$$
\begin{aligned}
Adjusted\ &Interest\ Spread_{i,t} \\
&= \alpha_0 + \sum_{k=1}^{8} \beta\ Bank\ Specific\ Variables_{k,it} \\
&+ \sum_{L=1}^{6} \beta\ Industry\ and\ Macroeconomic\ Variables_{L,it} \\
&+ \sum_{M=1}^{4} \beta\ Institutional\ Variables_{M,it} + \mu_i + \varepsilon_{it},
\end{aligned}
\tag{4}
$$

where $i = 1, 2, \ldots, n$, refers to banks, and $t = 1, 2, \ldots, T$, refers to yearly time period during the period 1988–2015. The $AIS_{it}$ is defined as the net interest margin for bank $i$ at year $t$; $\alpha_0$ represents the constant term. $u_i$ represents the time effects.

Following from prior literature, bank specific variables include the size of operations, credit risk, risk aversion, loan quality, operational inefficiency, liquidity risk, proportion of corporate and industrial loans, and local or foreign bank ownership.

Industry and macroeconomic variables include market concentration, market share, economic growth, economic growth inflation, real interest rate, and the proportion of national savings to the GDP.

Institutional variables include financial freedom distance, government spending distance, government effectiveness distance, and monetary freedom distance. The value $u_i$ denotes the individual effect, which does not change with time. The coefficient $\beta$ measures the sensitivity of each of these variables to the dependent variable.

We estimated four models respectively. Model 1 includes all bank specific variables. Model 2 includes industry and macroeconomic variables. Model 3 includes institutional variables. Model 4 is the full model with all the variables included.

In order to understand the institutional factors better, we examined whether institutional distance between China and other countries will weaken or eliminate the effects on the AIS. Therefore, we included an interaction term between each of the three institutional distances and the variable of local

or foreign bank with each of the bank-specific factors in the model. Following Balli and Sørensen (2013), the model used is specified below:

$$
\begin{aligned}
Adjusted\ Interest\ Spread_{i,t} \\
= \alpha_0 + \sum_{k=1}^{7} \beta\ Bank\ Specific\ Variables_{k,it} \\
+ \sum_{L=1}^{3} \beta\ Moderating\ Variables_{L,it} \\
+ \sum_{M=1}^{7} \beta(Moderator_{L,it} \\
- \overline{Moderator_{L,it}})(Bank\ Specific\ Variable_{K,it} \\
- \overline{Bank\ Specific\ Variable_{K,it}}) + \mu_i + \varepsilon_{it}.
\end{aligned}
\tag{5}
$$

## 5. Empirical Results

### 5.1. Summary Statistics

Table 2 provides the summary statistics on the variables used in the model.

This table shows the mean, standard deviation, minimum and maximum characteristics of all variables under consideration based on different groups of banks and on average. Overall, the mean value of the AIS for all the banks was approximately 2.967% and ranged from −1.30% to 11.125%.

Size of operations is proxied by the gross loans. Apparently, SOBs control substantial operations in the banking sector. The loan loss provision relative to total loans, which is an indicator of the quality of the credit portfolio, has an average value of 0.73%, which is consistent across different groups of banks in our sample. On average, risk aversion was around 9.185%, with foreign-owned banks having the highest average value of 20.691%, suggesting that they are well-capitalized banks. Liquidity risk is similar for all the banks with an average percentage of 34.216.

Notably, there is wide cross-bank variation in the sample with market power ranging from 18.945% for five large SOBs to near 0% for foreign incorporated banks in China.

China's average inflation rate over the study period was 3.08%, and GDP growth averaged 9.55%. Of the total loans, 76.366% went to the corporate and industrial sectors. Surprisingly, in a few cases the risk aversion ratios are negative. We found that four Chinese banks recorded negative equity in the years of 2004, 2005, 2006, 2007, and 2008 respectively. These four banks are Agricultural Bank of China (NY −727,605 million in 2008); Industrial and Commercial Bank of China (CNY −535,844 million in 2004); China Resources Bank of Zhuhai (CNY −514.1 million in 2007); China Everbright Bank (CNY −5490.4 million in 2004; CNY −2550.8 million in 2005, CNY −182.5 million in 2006).

The negative equity condition could be traced to the bank making losses year after year and borrowed to fund non-performing loans. Consequently, the liabilities exceeded assets resulting in a negative equity.

As the banking sector in China is largely government-owned, the negative equity was not considered to be a cause for worry as the government would recapitalize such banks out of the budget. In 1999, for example, to address the problem of non-performing loans, four asset management companies were established to take over the non-performing assets from the banks and sell them off to the investors. Despite being the major creditors in China, "Chinese banks seem to be unfairly neglected in the discussions on bankruptcy" (Wei and Chen 2018, p. 110).

**Table 2.** Summary Statistics.

| | Adjusted Interest Spread | Size of Operation (Millions CNY) | Credit Risk (%) | Risk Aversion (%) | Loan Quality (%) | Operational Inefficiency (%) | Liquidity (%) | Proportion of Corporate and Industrial Loan (%) | Market Share (%) | Market Concentration | Real Interest Rate (%) | GDP Growth Rate (%) | CPI (%) | National Gross Savings to GDP (%) | Government Effectiveness Distance | Government Spending Distance | Monetary Freedom Distance | Financial Freedom Distance |
|---|---|---|---|---|---|---|---|---|---|---|---|---|---|---|---|---|---|---|
| | | | | | | | Five Large State-Controlled Commercial Banks | | | | | | | | | | | |
| Mean | 2.630 | 3,020,800.000 | 0.684 | 5.197 | 9.354 | 1.475 | 22.717 | 79.550 | 10.393 | 1303.039 | 2.229 | 9.559 | 4.154 | 44.739 | 0.033 | 0 | 0 | 0 |
| Std. Dev. | 0.877 | 2,759,870.000 | 0.317 | 2.877 | 10.906 | 1.322 | 12.014 | 11.773 | 3.815 | 380.534 | 3.404 | 2.252 | 5.634 | 5.399 | 0.175 | 0 | 0 | 0 |
| Minimum | 0.628 | 18,549.500 | 0.055 | −13.710 | 0.860 | 0.467 | 6.576 | 36.494 | 2.668 | 420.632 | −7.977 | 3.907 | −1.408 | 36.459 | −0.349 | 0 | 0 | 0 |
| Maximum | 7.317 | 11,900,000.000 | 1.603 | 10.670 | 39.600 | 7.789 | 56.558 | 94.639 | 18.945 | 1794.081 | 7.348 | 14.231 | 24.237 | 51.966 | 0.408 | 0 | 0 | 0 |
| Count | 113 | 120 | 88 | 120 | 67 | 113 | 120 | 120 | 64 | 64 | 120 | 120 | 120 | 120 | 84 | 120 | 120 | 120 |
| | | | | | | | 33 Joint-Stock Commercial Banks | | | | | | | | | | | |
| Mean | 2.793 | 354,602.200 | 0.572 | 7.547 | 3.761 | 0.015 | 34.019 | 76.643 | 0.977 | 1295.152 | 2.057 | 9.693 | 3.644 | 46.779 | 0.067 | 0 | 0 | 0 |
| Std. Dev. | 1.120 | 548,796.800 | 4.102 | 7.933 | 9.675 | 0.014 | 45.204 | 14.216 | 0.985 | 369.168 | 3.284 | 2.128 | 4.755 | 4.994 | 0.160 | 0 | 0 | 0 |
| Minimum | −1.303 | 140.942 | −69.186 | −1.320 | 0.000 | 0.001 | 1.047 | 0.027 | 0.002 | 420.632 | −7.977 | 3.907 | −1.408 | 36.459 | −0.349 | 0 | 0 | 0 |
| Maximum | 9.215 | 2,824,286.000 | 6.901 | 64.800 | 99.300 | 0.115 | 738.464 | 98.816 | 3.683 | 1794.081 | 7.348 | 14.231 | 24.237 | 51.966 | 0.408 | 0 | 0 | 0 |
| N | 366 | 384 | 299 | 383 | 254 | 341 | 384 | 384 | 273 | 273 | 384 | 384 | 384 | 384 | 316 | 384 | 384 | 384 |
| | | | | | | | 79 Urban Commercial Banks | | | | | | | | | | | |
| Mean | 3.270 | 42,560.430 | 0.917 | 6.260 | 2.786 | 0.012 | 25.991 | 81.374 | 0.100 | 1301.168 | 1.988 | 9.666 | 2.680 | 48.801 | 0.089 | 0 | 0 | 0 |
| Std. Dev. | 1.309 | 69,869.060 | 0.682 | 2.398 | 6.919 | 0.004 | 11.524 | 11.785 | 0.134 | 365.298 | 2.687 | 2.075 | 2.237 | 3.590 | 0.138 | 0 | 0 | 0 |
| Minimum | 0.208 | 10.992 | −0.185 | −6.420 | 0.000 | 0.005 | 2.643 | 10.495 | 0.000 | 420.632 | −7.977 | 6.900 | −1.408 | 36.459 | −0.349 | 0 | 0 | 0 |
| Maximum | 11.125 | 775,390.000 | 5.863 | 23.590 | 100.000 | 0.035 | 71.378 | 98.853 | 0.009 | 1794.081 | 7.348 | 14.231 | 24.237 | 51.966 | 0.408 | 0 | 0 | 0 |
| Count | 767 | 786 | 701 | 786 | 559 | 670 | 777 | 786 | 717 | 717 | 786 | 786 | 786 | 786 | 756 | 786 | 786 | 786 |
| | | | | | | | 32 Rural Commercial Banks | | | | | | | | | | | |
| Mean | 3.240 | 57,826.340 | 0.948 | 6.656 | 3.697 | 0.012 | 28.859 | 81.219 | 0.118 | 1231.346 | 1.848 | 9.306 | 3.016 | 49.902 | 0.112 | 0 | 0 | 0 |
| Std. Dev. | 0.992 | 59,488.730 | 0.821 | 2.389 | 4.815 | 0.003 | 13.370 | 12.499 | 0.110 | 352.001 | 2.666 | 2.028 | 1.718 | 1.526 | 0.127 | 0 | 0 | 0 |
| Minimum | 1.184 | 1393.303 | −0.827 | 0.530 | 0.340 | 0.006 | 2.454 | 35.730 | 0.006 | 420.632 | −2.335 | 6.900 | −0.766 | 39.832 | −0.120 | 0 | 0 | 0 |
| Maximum | 5.936 | 297,325.700 | 6.620 | 12.220 | 22.990 | 0.023 | 64.169 | 97.933 | 0.365 | 1794.081 | 5.451 | 14.231 | 5.864 | 51.966 | 0.408 | 0 | 0 | 0 |
| Count | 165 | 170 | 147 | 170 | 108 | 136 | 170 | 170 | 169 | 169 | 170 | 170 | 170 | 170 | 170 | 170 | 170 | 170 |
| | | | | | | | 40 Foreign-Owned Banks | | | | | | | | | | | |
| Mean | 2.382 | 20,992.120 | 0.380 | 20.691 | 1.775 | 0.016 | 60.986 | 59.990 | 0.042 | 1233.020 | 2.075 | 9.258 | 2.920 | 49.416 | 1.469 | 1.643 | 0.768 | 5.265 |
| Std. Dev. | 1.005 | 29,208.750 | 1.084 | 16.780 | 7.464 | 0.008 | 70.852 | 23.065 | 0.054 | 379.234 | 2.899 | 2.001 | 2.497 | 2.953 | 0.565 | 3.001 | 0.764 | 3.423 |
| Minimum | 0.031 | 15.000 | −7.450 | 4.740 | 0.000 | 0.003 | 11.258 | 0.229 | 0.000 | 420.632 | −7.977 | 6.900 | −1.408 | 36.459 | −0.349 | 0.000 | 0.000 | 0.000 |
| Maximum | 7.417 | 174,758.500 | 7.616 | 94.710 | 79.890 | 0.060 | 897.637 | 97.773 | 0.269 | 1794.081 | 7.348 | 14.231 | 24.237 | 51.966 | 2.437 | 12.251 | 3.796 | 9.896 |
| Count | 329 | 330 | 311 | 332 | 216 | 280 | 329 | 332 | 312 | 312 | 332 | 332 | 332 | 332 | 321 | 332 | 332 | 332 |
| | | | | | | | Three Rural Cooperative Banks | | | | | | | | | | | |
| Mean | 4.709 | 21,312.510 | 1.277 | 10.644 | 1.120 | 0.016 | 29.251 | 95.019 | 0.025 | 1008.478 | 2.414 | 8.243 | 3.013 | 49.551 | 0.137 | 0 | 0 | 0 |
| Std. Dev. | 1.950 | 7617.206 | 0.270 | 0.667 | 0.486 | 0.003 | 8.045 | 2.890 | 0.008 | 263.470 | 2.628 | 1.222 | 1.362 | 0.817 | 0.153 | 0 | 0 | 0 |
| Minimum | 1.953 | 11,295.000 | 0.984 | 8.920 | 0.580 | 0.010 | 14.407 | 89.094 | 0.015 | 420.632 | −1.472 | 6.900 | 1.437 | 48.393 | 0.004 | 0 | 0 | 0 |
| Maximum | 7.180 | 33,225.300 | 1.726 | 11.230 | 2.060 | 0.019 | 39.065 | 97.639 | 0.033 | 1394.735 | 4.732 | 10.636 | 5.411 | 51.497 | 0.408 | 0 | 0 | 0 |
| Count | 10 | 10 | 6 | 10 | 9 | 8 | 10 | 10 | 10 | 10 | 10 | 10 | 10 | 10 | 10 | 10 | 10 | 10 |
| | | | | | | | Total 192 Banks | | | | | | | | | | | |
| Mean | 2.967 | 305,048.200 | 0.734 | 9.185 | 3.242 | 0.014 | 34.216 | 76.366 | 0.671 | 1276.889 | 2.024 | 9.548 | 3.062 | 48.321 | 0.352 | 0.303 | 0.141 | 0.970 |
| Std. Dev. | 1.227 | 1,055,552.000 | 1.949 | 7.915 | 7.474 | 0.009 | 40.189 | 17.040 | 2.231 | 368.914 | 2.908 | 2.084 | 3.277 | 4.074 | 0.615 | 1.436 | 0.443 | 2.514 |
| Minimum | −1.303 | 10.992 | −69.186 | −13.710 | 0.000 | 0.001 | 1.047 | 0.027 | 0.000 | 420.632 | −7.977 | 3.907 | −1.408 | 36.459 | −0.349 | 0.000 | 0.000 | 0.000 |
| Maximum | 11.125 | 11,900,000.000 | 7.616 | 94.710 | 100.000 | 0.115 | 897.637 | 98.853 | 18.945 | 1794.081 | 7.348 | 14.231 | 24.237 | 51.966 | 2.437 | 12.251 | 3.796 | 9.896 |
| Count | 1750 | 1800 | 1552 | 1801 | 1213 | 1548 | 1790 | 1802 | 1545 | 1545 | 1802 | 1802 | 1802 | 1802 | 1657 | 1802 | 1802 | 1802 |

Following Maudos and de Guevara (2004), we measured the degree of market concentration by HHI in each year for the 16-year period. As Figure 1 shows, the market concentration measured by HHI declined after 2007, indicating growing competition among China's banks. Yet, this competition did not exercise any significant influence on the AIS. The overall trend of the AIS is upward despite some fluctuations as can be seen from Figure 2. These two figures confirm the negative relationship between market concentration and the AIS. It suggests that interest rate liberalization did not per se help reduce net interest margins but forced the bank managements to change their existing business models following increased fluctuations in interest rates.

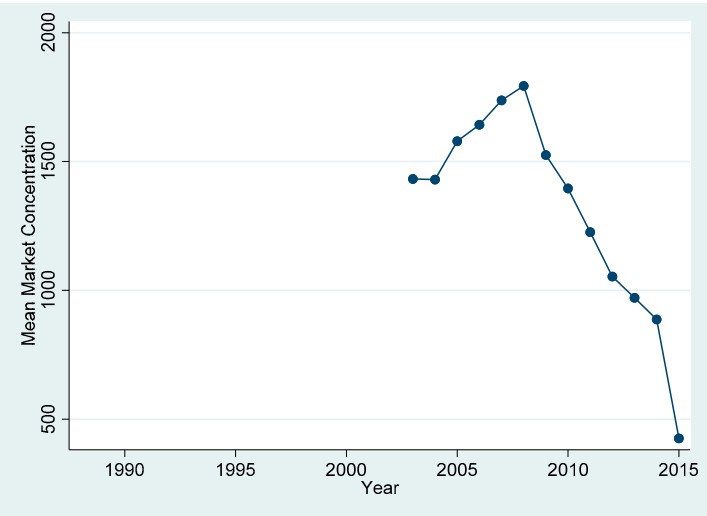

**Figure 1.** Mean market concentration of Chinese banks over the years until 2015.

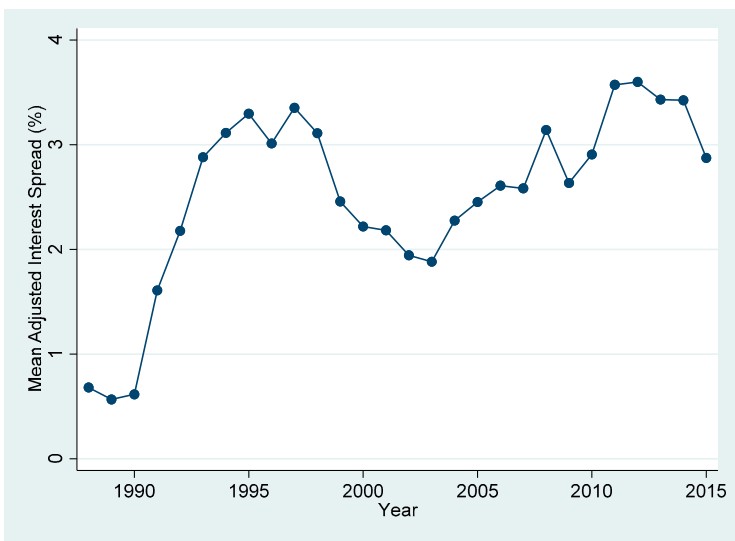

**Figure 2.** Mean adjusted interest spread of Chinese banks over the years until 2015.

*5.2. Correlation Coefficient between Variables*

Table 3 highlights the pair-wise correlations between all variables. It is noteworthy that the government effectiveness distance is highly correlated with financial freedom distance (0.8082), and foreign bank is highly correlated with financial freedom (0.8417), which is natural as foreign banks enjoy higher degree of financial freedom. Economic growth measured by GDP is highly correlated with market concentration (0.8174).

**Table 3.** Correlation coefficient between variables.

| | | (1) | (2) | (3) | (4) | (5) | (6) | (7) | (8) | (9) | (10) | (11) | (12) | (13) | (14) | (15) | (16) | (17) | (18) | (19) |
|---|---|---|---|---|---|---|---|---|---|---|---|---|---|---|---|---|---|---|---|---|
| AIS | (1) | 1 | | | | | | | | | | | | | | | | | | |
| Size of Operations | (2) | −0.059 * | 1 | | | | | | | | | | | | | | | | | |
| Credit Quality | (3) | 0.076 * | 0.073 * | 1 | | | | | | | | | | | | | | | | |
| Risk Aversion | (4) | −0.002 | −0.525 * | −0.206 * | 1 | | | | | | | | | | | | | | | |
| Loan Quality | (5) | −0.138 * | 0.037 | 0.081 * | −0.037 | 1 | | | | | | | | | | | | | | |
| Operational Inefficiency | (6) | 0.114 * | −0.151 * | −0.018 | 0.250 * | −0.006 | 1 | | | | | | | | | | | | | |
| Liquidity Risk | (7) | −0.050 * | −0.362 * | −0.156 * | 0.608 * | −0.085 * | 0.181 * | 1 | | | | | | | | | | | | |
| Proportion of Corporate and Industrial Loan | (8) | 0.349 * | 0.124 * | 0.017 | −0.247 * | 0.080 * | −0.076 * | −0.255 * | 1 | | | | | | | | | | | |
| Local or Foreign Bank | (9) | −0.227 * | −0.337 * | −0.085 * | 0.519 * | −0.084 * | 0.147 * | 0.308 * | −0.433 * | 1 | | | | | | | | | | |
| Market Share | (10) | −0.076 * | 0.587 * | −0.009 | −0.120 * | 0.116 * | −0.054 * | −0.099 * | 0.102 * | −0.138 * | 1 | | | | | | | | | |
| Market Concentration | (11) | −0.197 * | −0.203 * | −0.046 | −0.045 | 0.108 * | 0.082 * | −0.017 | 0.104 * | −0.060 * | 0.033 | 1 | | | | | | | | |
| Real Interest Rate | (12) | −0.009 | 0.080 * | 0.015 | 0.031 | 0.038 | −0.035 | −0.012 | −0.007 | 0.012 | −0.017 | −0.558 * | 1 | | | | | | | |
| GDP Growth Rate | (13) | −0.183 * | −0.168 * | −0.040 | −0.055 * | 0.093 * | 0.022 | −0.017 | 0.040 | −0.070 * | 0.030 | 0.817 * | −0.570 * | 1 | | | | | | |
| Inflation | (14) | 0.105 * | −0.037 | 0.005 | 0.002 | −0.141 * | 0.173 * | 0.051 * | −0.099 * | −0.040 | −0.016 | 0.269 * | −0.724 * | 0.327 * | 1 | | | | | |
| National Gross Savings to GDP | (15) | 0.155 * | 0.003 | 0.008 | 0.010 | −0.331 * | −0.176 * | −0.004 | 0.036 | 0.142 * | −0.069 * | 0.422 * | −0.312 * | 0.208 * | 0.008 | 1 | | | | |
| Government Effectiveness Distance | (16) | −0.199 * | −0.247 * | −0.087 * | 0.357 * | −0.111 * | 0.135 * | 0.235 * | −0.444 * | 0.878 * | −0.120 * | −0.021 | −0.015 | −0.038 | −0.020 | 0.153 * | 1 | | | |
| Government Spending Distance | (17) | −0.171 * | −0.184 * | −0.057 * | 0.269 * | −0.052 | 0.107 * | 0.1847 * | −0.334 * | 0.460 * | −0.065 * | −0.019 | −0.008 | −0.024 | −0.010 | 0.075 * | 0.356 * | 1 | | |
| Monetary Freedom Distance | (18) | −0.195 * | −0.119 * | −0.060 * | 0.186 * | −0.093 * | 0.145 * | 0.129 * | −0.260 * | 0.697 * | −0.095 * | −0.016 | 0.003 | −0.025 | −0.026 | 0.152 * | 0.734 * | 0.186 * | 1 | |
| Financial Freedom Distance | (19) | −0.163 * | −0.259 * | −0.060 * | 0.347 * | −0.101 * | 0.145 * | 0.232 * | −0.391 * | 0.842 * | −0.115 * | −0.078 * | 0.013 | −0.080 * | −0.028 | 0.131 * | 0.808 * | 0.336 * | 0.599 * | 1 |

* Correlation is significant at the 0.05 level (two-tailed).

The high national gross savings rate (the proportion of national savings to the GDP) comes from China' high trade surpluses, which results in an economy driven by investment rather than domestic consumption. Bosworth (2014) pointed out that higher national savings rate relative to domestic investment will reduce domestic interest rates.

Another factor behind the negative relationship is that China controls its interest rate, and investors have very limited financial instruments to invest. Hung and Qian (2013) suggest that emerging economy like China will keep real interest rates low to force the national saving rate to rise in order to provide cheap credit to industries.

The correlation coefficient is further checked with the variance inflation factor (VIF) when performing the regression analysis procedures for Model 4. The VIF signifies the degree to which each independent variable is explained by the other independent variable, and all variables are found well below the suggested cut-off point of 10 (Hair et al. 1998). The variable of local or foreign bank was found to be highly correlated to the other independent variables and was omitted from analysis.

*5.3. Fixed Effects Regression Results for AIS*

We interpret the results based on the fixed effects regression analysis for all the models. The results are exhibited in Tables 4–7.

5.3.1. Bank Specific Factors and AIS

Results from Models 1 and 4 (refer to Table 4) show that four of the bank specific factors are important determinants of the AIS. Loan loss provisions over total loans is a measure of a bank's credit quality. The coefficient of this variable is positive and significant which indicates that higher loan loss provision leads to higher AIS. This result is consistent with Poghosyan (2012) that banks are expected to require higher interest margins to compensate for funding riskier projects, and to maintain adequate loan reserves. Marinković and Radović (2014) note 'at least for existing loan customers, if possible, banks will re-price existing loans as they become due or when they are renegotiated. This will generate a positive relation between default risk and the NIM, ceteris paribus'.

The coefficient of risk aversion is positive and significant, suggesting that Chinese banks are imposing an extra bank interest margin as a compensation for taking systematic risk. Similar results are reported by Ho and Saunders (1981) and Williams (2007) for the Australian banking sector. Meanwhile, operational inefficiency has a positive relationship with the AIS (marginally at 10%), indicating banks with higher operating cost will have higher AIS than banks with lower operating costs by providing their intermediation services. Birchwood et al. (2017) have similar findings in the context of banks in Central America and the Caribbean.

The variable of liquidity risk is significant and negatively associated with the AIS. This result is not consistent with the literature that banks with higher risk on their credit books are likely to adjust upwards their interest margins to cover expected losses arising from default, when compared to the banks with lower credit risk (Angbazo 1997; Mody and Peria 2004). It is also contradictory with the results from Birchwood et al. (2017). The result indicates that Chinese banks holding large liquid assets to meet either regulatory requirements or depositors' withdrawals, failed to price this factor in their interest margins. The SOBs also get flushed with large cash deposited by SOEs unless they can find avenues for interbank lending.

**Table 4.** Fixed effects regression results: adjusted interest spread (AIS) as the dependent variable.

| Variables | (1) | (2) | (3) | (4) |
|---|---|---|---|---|
| | Model 1 | Model 2 | Model 3 | Model 4 |
| *Bank Specific Variables* | | | | |
| Size of Operations | 0.0453 | | | 0.0598 |
| | (0.107) | | | (0.108) |
| Credit Quality | 11.64 *** | | | 12.42 *** |
| | (4.017) | | | (3.998) |
| Risk Aversion | 0.0477 *** | | | 0.0466 *** |
| | (0.00977) | | | (0.00993) |
| Loan Quality | 0.00110 | | | −0.000120 |
| | (0.00254) | | | (0.00141) |
| Operational Inefficiency | 30.57 * | | | 30.18 * |
| | (15.67) | | | (15.80) |
| Liquidity Risk | −0.0106 *** | | | −0.0105 *** |
| | (0.00199) | | | (0.00200) |
| Proportion of Corporate and Industrial Loans | 1.550 *** | | | 1.658 *** |
| | (0.283) | | | (0.265) |
| *Industry and Macro-economic Variables* | | | | |
| Market Share | | 2.904 | | 2.792 |
| | | (2.613) | | (2.222) |
| Market Concentration | | −0.000210 | | −0.000720 |
| | | (0.000473) | | (0.000457) |
| Real Interest Rate | | 0.119 | | 0.170 |
| | | (0.155) | | (0.140) |
| GDP Growth Rate | | 0.696 | | 1.217 *** |
| | | (0.433) | | (0.414) |
| Inflation | | −0.0668 | | −0.170 *** |
| | | (0.0467) | | (0.0523) |
| Proportion of national savings to the GDP | | 0.221 ** | | 0.243 *** |
| | | (0.0858) | | (0.0772) |
| *Institutional Variables* | | | | |
| Government Effectiveness Distance | | | 0.0964 | −0.0184 |
| | | | (0.152) | (0.142) |
| Government Spending Distance | | | −0.205 | 0.000996 |
| | | | (0.146) | (0.0848) |
| Monetary Freedom Distance | | | −0.107 | −0.126 |
| | | | (0.105) | (0.0990) |
| Financial Freedom Distance | | | 0.000892 | −0.0542 |
| | | | (0.0353) | (0.0411) |
| Constant | 0.210 | −12.91 ** | 2.810 *** | −19.82 *** |
| | (0.814) | (5.004) | (0.0948) | (5.388) |
| Observations | 1206 | 1206 | 1206 | 1206 |
| Adjusted R-Squared | 0.491 | 0.363 | 0.363 | 0.494 |
| Number of Banks | 172 | 172 | 172 | 172 |

Robust standard errors in parentheses. *** $p < 0.01$, ** $p < 0.05$, * $p < 0.1$.

**Table 5.** The moderating effect of government effectiveness and the impact of AIS.

| Variables | Model 1 |
|---|---|
| ***Bank Specific Variables*** | |
| Size of Operations | −0.0424 |
| | (0.135) |
| Credit Quality | 11.44 *** |
| | (4.118) |
| Risk Aversion | 0.0365 *** |
| | (0.00912) |
| Loan Quality | 0.00243 |
| | (0.0212) |
| Operational Inefficiency | 33.38 *** |
| | (12.66) |
| Liquidity Risk | −0.00908 *** |
| | (0.00191) |
| Proportion of Corporate and Industrial Loans | 1.659 *** |
| | (0.271) |
| ***Moderator*** | |
| Government Effectiveness Distance | 2.683 |
| | (8.815) |
| ***Interaction Term*** | |
| Government Effectiveness Distance × Size of Operations | −0.110 |
| | (0.0935) |
| Government Effectiveness Distance × Credit Quality | −2.243 |
| | (3.754) |
| Government Effectiveness Distance × Risk Aversion | −0.00731 |
| | (0.00601) |
| Government Effectiveness Distance × Loan Quality | 0.000908 |
| | (0.0213) |
| Government Effectiveness Distance × Operational Inefficiency | 4.224 |
| | (7.880) |
| Government Effectiveness Distance × Liquidity Risk | 0.00179 |
| | (0.00134) |
| Government Effectiveness Distance × Proportion of Corporate and Industrial Loans | −0.427 *** |
| | (0.153) |
| Constant | −1.686 |
| | (8.535) |
| Observations | 1206 |
| Number of Banks | 172 |
| Adjusted R-Squared | 0.501 |

Robust standard errors in parentheses. *** $p < 0.01$.

**Table 6.** The moderating effect of financial freedom and the impact of AIS.

| Variables | Model |
| --- | --- |
| ***Bank Specific Variables*** | |
| Size of Operations | 0.0390 |
| | (0.112) |
| Credit Quality | 12.30 *** |
| | (4.314) |
| Risk Aversion | 0.0391 *** |
| | (0.0120) |
| Loan Quality | 0.00491 |
| | (0.00532) |
| Operational Inefficiency | 30.17 * |
| | (16.17) |
| Liquidity Risk | −0.0101 *** |
| | (0.00196) |
| Proportion of Corporate and Industrial Loans | 1.836 *** |
| | (0.311) |
| ***Moderator*** | |
| Financial Freedom Distance | 0.134 |
| | (2.891) |
| ***Interaction Term*** | |
| Financial Freedom Distance × Size of Operations | −0.0347 |
| | (0.0306) |
| Financial Freedom Distance × Credit Quality | −1.313 |
| | (0.993) |
| Financial Freedom Distance × Risk Aversion | −0.00275 |
| | (0.00259) |
| Financial Freedom Distance × Loan Quality | 0.00413 |
| | (0.00588) |
| Financial Freedom Distance × Operational Inefficiency | 1.284 |
| | (2.642) |
| Financial Freedom Distance × Liquidity Risk | 0.00110 * |
| | (0.000586) |
| Financial Freedom Distance × Proportion of Corporate and Industrial Loans | −0.119 * |
| | (0.0697) |
| Constant | 0.118 |
| | (2.816) |
| Observations | 1206 |
| Number of Banks | 172 |
| Adjusted R-Squared | 0.499 |

Robust standard errors in parentheses. *** $p < 0.01$, * $p < 0.1$.

**Table 7.** The moderating effect of monetary freedom and the impact of AIS.

| Variables | Model 1 |
|---|---|
| ***Bank Specific Variables*** | |
| Size of Operations | −0.0185 |
| | (0.175) |
| Credit Quality | −1.874 |
| | (5.428) |
| Risk Aversion | 0.0479 *** |
| | (0.0104) |
| Loan Quality | 0.0123 |
| | (0.0213) |
| Operational Inefficiency | 38.65 *** |
| | (13.24) |
| Liquidity Risk | −0.00120 |
| | (0.00341) |
| Proportion of Corporate and Industrial Loans | 1.225 *** |
| | (0.354) |
| ***Moderator*** | |
| Monetary Freedom Distance | −5.026 |
| | (17.70) |
| ***Interaction Term*** | |
| Monetary Freedom Distance × Size of Operations | −0.0798 |
| | (0.132) |
| Monetary Freedom Distance × Credit Quality | −15.81 ** |
| | (7.007) |
| Monetary Freedom Distance × Risk Aversion | 0.00498 |
| | (0.0153) |
| Monetary Freedom Distance × Loan Quality | 0.0113 |
| | (0.0221) |
| Monetary Freedom Distance × Operational Inefficiency | 11.03 |
| | (16.25) |
| Monetary Freedom Distance × Liquidity Risk | 0.0107 *** |
| | (0.00390) |
| Monetary Freedom Distance × Proportion of Corporate and Industrial Loans | −0.602 * |
| | (0.355) |
| | (0.373) |
| Constant | 5.573 |
| | (17.79) |
| Observations | 1206 |
| Number of Banks | 172 |
| Adjusted R-Squared | 0.499 |

Robust standard errors in parentheses. *** $p < 0.01$, ** $p < 0.05$, * $p < 0.1$.

Results from Models 1 and 4 also reveal that banks with a higher proportion of corporate and industrial loans generate higher AIS. These results are quite consistent with the condition of lending and deposits rate policies in China. These assets comprised mostly of financial securities. Furthermore, deposit interest rate in China averaged 1.16% from 1990 until 2018, reaching an all-time high of 3.15% in July of 1993 and a record low of 0.35% in July of 2012 (Trading Economics 2012). The low deposit rate would have resulted in higher AIS.

5.3.2. Industry and Macroeconomic Variables and AIS

Regarding the industry and macro-economic factors, Models 3 and 4 show positive and significant effects of GDP growth rate, and the proportion of national savings to the GDP to the AIS. It could be because Chinese banks can take deposits from households with an artificially low level of interest and

transfer or subsidize the corporate sector, particularly the SOEs. Bank performance was found to be positively related to the overall economic development in China which is consistent with the theory that high net interest margin is associated with a high rate of GDP especially in developing countries (Gelos 2009; Tan and Floros 2012).

### 5.3.3. Institutional Variables and AIS

Finally, the level of difference in government spending, and monetary freedom between home country China and foreign countries are not significant as can be seen from Model 3 in Table 4. Over the years, Chinese governments have been borrowing heavily and the related government securities such as complex off-balance-sheet structures are developed by banks to avoid tightening regulations that restrict bank loans to local governments (Wu 2015). Hence, some of these assets appear as investments rather than loans on the balance sheet, and apparently these investments failed to produce good return to banks.

Results from Model 3 reveal that the level of difference in monetary freedom between home country China and foreign countries is not significant. Over the years, both central and local Chinese governments borrowed funds to finance government projects. Banks invest in treasury bills which are issued by the government to raise short-term debt. Domestic banks have a tradition that they would lend the money to SOEs and the government rather than to the private sector for various reasons. Moreover, both the government and the SOEs would receive cheaper finance to fund their projects. Treasury bills being risk free typically pay much lower interest rate than the market rate which in turn could impact the bank interest margin.

The time-invariant variable of foreign bank or local bank was omitted from the fixed-effect model due to collinearity. In order to understand the relationship between this independent variable and the AIS, we adopted the approach of Pesaran and Zhou (2018) to identify and estimate the effects of this time-invariant variable. It yielded the following results: coefficient of the variable ($-0.583$), standard error ($0.320$), and $t$-value ($-1.82$). These estimations indicate that foreign banks operating in China have lower AIS compared to local Chinese banks. This result is generally consistent with Williams (2007) in the context of the Australian banking sector.

Building upon the Models 3 and 4 in Tables 4–7 report the results of the interaction of financial freedom distance, monetary freedom distance, and government effectiveness distance with each of the bank specific variables.

Results from Tables 5–7 show that the coefficients of financial freedom distance, monetary freedom distance, and government effectiveness distance variables are not significant. Results from Tables 5–7 also show that these three institutional variables interact with credit quality, liquidity risk, and proportion of corporate and industrial loans leading to lower AIS. These results suggest that when the distance in institutional factor is high, the expected and potential positive relationship between the AIS and credit risk and proportion of corporate and industrial loan will be strengthened. These results support that institutional factors have a significant moderating effect on the AIS.

### 5.4. Robustness Tests

To guarantee the robustness of our empirical results, we conducted robustness tests using various methods. First, we replaced our dependent variable with net interest margin and interest spread. Second, we dropped the data before 2006 (due to lack of observations on some of the variables) and focused on the sample data between 2007 and 2015. Third, we grouped the data based on the size of the banks, namely, large and small.

These robustness tests confirm our main findings in Tables 4–7, thus indicating that our results are robust. The robustness test results are available upon request from the authors.

## 6. Conclusions

The aim of this paper was to develop our understanding of the determinants of bank interest margins in China. Prior studies are generally confined to western countries. In China, the state-owned-banks dominate the sector. Consequently, China provides a unique context for the study. The data of Chinese banks over the period 1988–2015 were analyzed.

The study found that five bank specific variables—credit quality, risk aversion, liquidity risk, and proportion of corporate and industrial loans—significantly affected the adjusted interest spread. We found that macro-economic variables such as GDP growth rate, inflation, and the proportion of national savings to the GDP have significant association with the AIS. None of the institutional variables had any significant association with the AIS.

The bank competition in China has increased after the (WTO) agreement, and the relaxation of policies applied to foreign banks operating in China. Yet, the AIS or net interest margin for Chinese banks are not under pressure following higher foreign counterparts' competition. This scenario is in divergence with the literature. Overall, China's banking sector is still operating in a relatively restricted environment, and free market pricing mechanism is not yet present. The government is subsidizing the stated owned enterprises by offering a lower deposit rate.

**Author Contributions:** Writing-Original Draft Preparation, L.X.L.; Writing-Review & Editing, M.S.

**Conflicts of Interest:** The authors declare no conflict of interest.

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
