# Peer review of "Bank Interest Rate Margin, Portfolio Composition and Institutional Constraints"

_jrfm, doi:10.3390/jrfm12030121_

Round 1

Reviewer 1 Report

Please find my review in the attachment.

Reviewer 2 Report

This is an interesting paper discussing the factors affecting the adjusted interest spread for banks in China, where the interest rates had been heavily regulated. However, I have following comments and suggestions:

 1. In this paper, leverage is measured by the ratio of total loans to customer deposits(LDR). This is not a measure for leverage but for liquidity. ratio less than 1 means the bank holds some cash from deposit as contingencies (80%-90% is common for a traditional commercial bank). The measure for risk aversion, i.e. ratio of total equity to total assets, is the typical measure for leverage. 

2. Table 2 for summary statistics is hard to read. It appears some variables are presented in percentage while others are in ratio. Please make it consistent. Are AIS, leverage and risk aversion in percentage? Are proportion of C&I loans, 4 contribution variables, and market share in ratio? What's the unit for "size of operation", USD or CNY, millions or billions? 

3. The market share in Table 2 does not make sense. It is said the mean market share for large state-owned bank is "0.059" (5.9%?) in the table. However, on line 105, it is said that 85.8% of total bank assets come from 5 largest state controlled commercial banks. 

4. Policy banks are very different from commercial banks and don't take deposits. They are policy oriented and not for profit. They are not regulated the same way as commercial banks. So analyzing theirs AIS can be problematic. It raised my concern when I saw the over 200% loan to deposit ratio (or "leverage" in this paper) before I went to find  more about policy banks in China. So please at least describe the difference between policy bank and commercial bank in the context, or delete them from the data.

5. four variables for contribution of assets/liability to AIS are not appropriate in this paper. the authors do not understand the cited literature, Covas et al. (2015). These four measures are appropriate in the original paper as they are used to explain the change in banks' NIM through their decomposition into the change in yields and portfolio composition (with respect to different asset/liability class).Given definition described in the paper, contribution to AIS of loan income = (interest income from loans/total interest earning assets)*((loans/total interest earning assets)= interest income from loans/total interest earning assets. Assume banks with only loans and securities as interest earning assets, 1% of contribution of loan income to total earning assets can have very different impact for a bank with the  4% or 10% total interest income (as percentage of total earning assets) 

6. In term of regression technique, please report robust standard errors to control for potential heteroscedasticity and serial correlation as the t-test can be biased using regular standard errors. Meanwhile, as FE regression has been identified by Haussman test, there is no need to show RE regression results.

Round 2

Reviewer 1 Report

Please find my review in the attachment.

Reviewer 2 Report

The paper has improved a lot since last version. But I have the following comments.

Credit quality and loan quality in this version of the paper are actually measuring the same thing though from slight different aspects. Normally the two terms are used interchangeably in the banking literature. In this paper, two variables are included in the regressions. The credit quality is loan provision/total loans, a short-term effect on income statement, while loan quality is NPL/total loans, a long-term effect from balance sheet. Meanwhile, Section 4.2.1 suggests credit quality has a negative impact and loan quality has a positive impact, while the regressions suggest the opposite. So it worths at least more discussions about the conflict signs from the empirical results if both variables are included in the regression. The conflict signs could be due to the potential endogeneity problem between these two variables.
